# Quantifying hot topic dynamics in scientific literature: An information-theoretical approach

**Artem Chumachenko***

Centre for European Regional and Local Studies (EUROREG), Science Studies Laboratory, University of Warsaw, Warsaw, Poland

* a.chumachenko@uw.edu.pl

## Abstract

Understanding the internal structure of scientific discourse is essential for tracking the evolution of research topics and their conceptual interdependencies. However, existing approaches such as dynamic topic modeling and neural topic models often fail to capture fine-grained semantic shifts among known concepts, or require substantial computational resources. Co-occurrence networks offer a more interpretable alternative, but typically rely on correlation-based weights that lack metric properties, preventing rigorous temporal comparison and topological interpretation.

To address this gap, we introduce a metric-based framework for analyzing the evolving structure of concept networks in the scientific literature. Using 10,370 research articles (2010-2023) on international security from JSTOR and PORTICO, we compute the normalized variation of information (NVI) distances to construct annual concept networks with a well-defined geometric structure. We then quantify semantic change using velocity matrices and extract major trends using Minimum Spanning Tree (MST) analysis.

Our results reveal that conceptual shifts are concentrated in temporally localized hubs and are not driven by co-occurrence frequency alone, but by contextual information and shared uncertainty between concept distributions. By introducing a scalable, interpretable, and mathematically grounded approach to tracking concept dynamics, this study contributes new tools for topic evolution analysis and offers insight into the structural organization and reconfiguration of knowledge over time.

## Introduction

Rapid accumulation of textual data—from scientific literature and policy documents to social media and news—has driven extensive research into the dynamics of topic evolution. Advances in computational linguistics and machine learning have facilitated the development of models capable of identifying and tracking emerging trends over time. Among these, dynamic topic modeling (DTM) [1] has provided valuable information on how themes emerge, peak, and decline, with applications in diverse disciplines. For example, [2] applies DTM to analyze traffic-related discussions, while [3] explores its role in monitoring the

processed data are available at https://doi.org/10.5281/zenodo.14948128 as Mathematica .m files. Codes to reproduce figures are available on Github (https://github.com/ArtemEsper/TopicGeometry).

**Funding:** The author(s) received no specific funding for this work.

**Competing interests:** The authors have declared that no competing interests exist.

exponential growth of academic literature. Roberts et al. [4] introduced Structural Topic Modeling (STM), which embeds document-level metadata directly into the generative process and has been widely used in sociology, political science, and science studies to correlate topics with covariates. Studies such as [5,6] use STM to examine policy discourse or media framing. In contrast to the probabilistic weighting of the STM, our NVI network approach offers a purely metric-based representation of the concept distances, facilitating explicit topological analyses without assuming Dirichlet priors. Beyond topic modeling, hybrid approaches integrating static and dynamic models have been used to examine the temporal and hierarchical structure of document collections [7]. In addition, context-sensitive models such as BERTopic have been used to track the evolution of public discourse, as demonstrated in [8], which analyzed the COVID-19 vaccination debates in Brazil. More advanced technologies such as neural topic models (NTMs) offer even better scalability and flexibility without requiring model-specific parameter derivations [9–11]. These developments underscore the increasing importance of computational techniques in capturing the interplay between external events and evolving research themes.

Recent work has also highlighted the persistent challenges in evaluating topic models, particularly with respect to coherence and human interpretability. Tan and D'Souza (2025) propose using large language models (LLMs) to bridge this gap, demonstrating that LLMs can serve as effective evaluators of topic quality in diverse corpora [12].

Despite these advancements, the practical implementation of these technologies remains challenging. One of the primary obstacles is computational complexity – sequential dependencies between time periods in DTM introduce additional parameters, significantly increasing processing time and memory requirements, particularly for large-scale corpora [13]. Additionally, data sparsity in specific time slices can lead to unstable topic distributions, making it difficult to track meaningful trends, especially for emerging topics with limited historical presence. Another key challenge is topic drift and instability, where changes in word usage may reflect shifts in vocabulary rather than true conceptual evolution. Ensuring consistency in topic trajectories over time often requires post-processing techniques, such as topic alignment or embedding-based transformations. Furthermore, determining the optimal granularity for temporal segmentation is non-trivial: narrow time windows introduce excessive noise, while broad intervals obscure significant conceptual transitions. Overcoming these limitations requires hybrid methodologies that integrate neural topic models, concept co-occurrence networks, and embedding-based tracking to improve model robustness and interpretability.

Although DTM and NLM approaches are powerful in discovering new topics, there are many scenarios where researchers or practitioners are more concerned with examining the internal dynamics of a known topic. This is particularly relevant in fields where a core vocabulary has already been established [14,15]. In such cases, the primary objective is not to identify hidden structures across multiple topics but rather to analyze how relationships between established concepts evolve over time.

To meet the requirements of smaller corpora, lower computational complexity, and improved interpretability, co-occurrence networks provide a compelling alternative to topic modeling approaches. These networks have long been used as a versatile tool for studying conceptual evolution within a well-defined domain. They have been widely applied in fields such as ecology and microbiome research [16,17], bibliometrics [18], information science [19], artificial intelligence [20], and scientific discourse analysis [21]. By representing nodes (e.g., species, concepts, or terms) and edges (the strength of associations between them), co-occurrence networks enable researchers to examine interactions that may reflect underlying conceptual relationships [22,23].

The same principles have been successfully applied to concept co-occurrence in text corpora, where these networks provide a structured representation of how terms and ideas interrelate within a given domain. Unlike topic modeling, which often produces high-level topic distributions, co-occurrence networks capture direct contextual relationships between concepts, making them a valuable framework for tracking the evolution of conceptual structures in the scientific literature [21,24]. Their ability to highlight key associations within a predefined conceptual space makes them especially useful for researchers interested in tracing the evolution of knowledge, detecting semantic shifts, and analyzing field-specific discourse trends without the overhead of complex probabilistic models.

The construction of co-occurrence networks requires appropriate weighting methods for edges, as these directly influence the interpretation of network topology. Correlation-based metrics, such as Spearman's or Pearson's correlation, are commonly used to generate co-occurrence matrices from text or abundance data, forming the foundation of network edges [25,26]. However, these traditional methods often struggle to capture higher-order dependencies in complex networks. To address this, more advanced techniques such as Weighted Correlation Network Analysis (WCNA) help identify hub concepts by leveraging topological overlap measures, which evaluate shared connections between nodes [27, 28]. Additionally, SparCC (Sparse Correlations for Compositional Data) has been developed to handle the unique challenges posed by compositional datasets, ensuring a more robust estimation of associations in cases where standard correlation measures may be unreliable [29].

Despite their utility, commonly used co-occurrence network weighting methods often lack metric properties, such as the triangle inequality, which is fundamental in many mathematical and geometric analyses [30]. The absence of a proper metric structure limits the applicability of quantitative techniques that rely on well-defined distance measures and restricts the rigorous mathematical interpretation of node relationships.

Incorporating a metric that satisfies the triangle inequality for data that reside in the information space provides a more structured and interpretable representation of distances within the network. Specifically, if two nodes are close to a central node, they cannot be very far from each other, ensuring a coherent and logically consistent representation of conceptual proximity in information networks. This property improves network topology analysis by improving the detection of clusters, hierarchical structures, and interaction patterns. In contrast, alternative similarity measures often lack this geometric intuition, leading to less interpretable and potentially misleading network structures.

Although generalizations of geometric and topological data analysis (TDA) methods have been developed to extend Euclidean approaches to Bregman geometries, these methods require specialized adaptations [31]. Their integration into existing network analysis frameworks remains challenging, as they often demand customized computational tools to align with Euclidean-based methodologies. By incorporating metric-based approaches into co-occurrence network analysis, researchers can achieve greater robustness and precision, enabling more accurate insights into community structures and the evolution of conceptual relationships.

To address the limitations of traditional co-occurrence-based similarity measures, we propose an approach that uses the normalized Variation of Information (NVI) metric to compute pairwise distances between concepts within a single known topic and track how these distances evolve over time. NVI is a well-defined metric that satisfies the triangle inequality, ensuring a mathematically rigorous and consistent measure of conceptual similarity and divergence [32]. Unlike simpler similarity measures that rely purely on co-occurrence

frequencies, NVI quantifies the shared and unique information between concept distributions, making it particularly suited for detecting subtle semantic shifts.

Originally developed for clustering validation and hierarchical clustering analysis, NVI has been widely applied in information-theoretic comparisons of partitions and dynamic network analysis, providing a robust framework for monitoring evolving relationships in dynamic datasets [33–35]. Using these properties, our approach offers a principled method for capturing conceptual drift, allowing for a structured and interpretable analysis of the evolution of the topic over time.

For each consecutive time period, we construct a distance matrix that captures contextual differences between concepts based on their distributional properties. By analyzing timely changes in these matrices, we derive "velocity matrices", which quantify how quickly conceptual relationships evolve – a perspective that complements existing topic modeling strategies. Unlike traditional methods that rely solely on co-occurrence frequencies, VI-based distances account for the uncertainty and information loss between concept distributions, offering a more nuanced representation of conceptual shifts.

This methodology allows for the identification of "fast-moving" concepts whose relationships with others undergo the most dramatic structural shifts, thereby providing a fine-grained view of topic evolution beyond the aggregated signals that conventional modeling might overlook. Using VI's metric properties, we ensure logical consistency in measuring conceptual drift, leading to more interpretable and robust insights into the dynamics of the knowledge network.

Although our approach intersects with topic modeling and co-occurrence-based methods, it is not intended to replace or outperform them in identifying latent topic structures across large corpora. Instead, we propose a complementary method that aims at analyzing the internal dynamics of a known topic, where a core conceptual vocabulary is already defined. Our goal is not topic detection, but tracking how relationships between selected concepts evolve over time, a question that traditional topic modeling frameworks often leave unaddressed. We believe that our metric-based approach can be used in conjunction with models like LDA or MDS: topic models can identify broad thematic areas, while our method can further analyze within-topic structure, conceptual convergence/divergence, and the influence of knowledge producers on semantic dynamics across time.

We apply our metric-based approach to a dataset of 10,370 research articles (2010–2023) related to "International Relations" from JSTOR and PORTICO, constructing evolving metric-based co-occurrence networks to quantify how conceptual relationships evolve over time. Our analysis reveals that:

- By computing velocity matrices and applying Minimum Spanning Tree (MST) analysis, we visualize the drift of concepts over time and identify localized hubs where conceptual shifts tend to concentrate, highlighting areas of emerging research focus.
- These networks provide insights into how key concepts interact, how certain ideas become central within the knowledge structure, and how conceptual relationships are reconfigured as academic discourse evolves.

Our results support the idea that dynamic knowledge structures are anchored in stable conceptual hubs, with major shifts occurring at specific points of high interaction, where conceptual relationships are actively reshaped over time.

The remainder of this paper is structured as follows. Section **Materials and Methods** provides a description of the dataset, detailing the data sources, pre-processing steps, and transformations applied for analysis. The model description outlines the methodology behind

our Variation of Information (VI)-based distance matrices and velocity matrix computations. The **Results** section presents the application of our metric-based co-occurrence network approach to the topics of cybersecurity, international security, international relations, human rights, and sanctions, highlighting key conceptual changes and structural patterns. Finally, we conclude with future research directions and discuss potential extensions of this approach for broader scientometric and knowledge network applications.

## Materials and methods

### Database description

For our analysis, we used a corpus of 10,370 English-language documents obtained through JSTOR's Constellate platform (http://constellate.org), published between 2010 and 2024, with a primary focus on International Security. Each JSTOR record includes the full text of the document along with associated metadata, such as title, authors, publisher, publication date, and the frequencies of all unigram, bigram, and trigram occurrences in the title, abstract, and main text.

To construct a dictionary of concepts, we extracted keyword sections across all documents and identified the most frequent unique n-grams of each type, yielding a vocabulary of 46,287 keywords. A Google BigQuery-based pipeline was used for data cleaning and duplication, ensuring high-quality document-concept-frequency mappings. The processed dataset was then stored in a transactional MySQL database for further analysis.

### Model description

We use an information-theoretic metric called *normalized Variation of Information (NVI)*, which was originally introduced to measure the distance between partitions (or 'clusters') in cluster analysis [32,36–38]. In our context, rather than comparing clusters, we apply NVI to quantify how different the *frequency distributions* of two concepts are. Specifically, let $X$ and $Y$ be random variables that represent the frequency distributions of two distinct concepts in a set of documents. By interpreting these distributions as analogous to clusters, NVI provides an intuitive measure of how far apart the concepts are in terms of their observed frequencies and co-occurrence patterns. The metric is defined as[33]:

$$d(X,Y) = nVI(X,Y) = \frac{VI(X,Y)}{H(X,Y)} = \frac{H(X) + H(Y) - 2M(X,Y)}{H(X,Y)} = 1 - \frac{M(X,Y)}{H(X,Y)}. \tag{1}$$

Here, $M(X,Y)$ denotes their *mutual information*, and $H(X,Y)$ together with $H(X)$ and $H(Y)$ are the *joint* and *marginal* Shannon entropies, respectively.

The metric $d(X,Y)$ at any given $t$ is a proper distance metric, satisfying the identity of indiscernibles, symmetry, and the triangle inequality [32]. If two concepts share the same frequency distribution across the corpus, then $d(X,Y) = 0$; if they never appear together in any document, $d(X,Y) = 1$. The properties of $d$ as a metric are intuitive and robust for comparison. The triangle inequality, in particular, implies that if two concepts (or clusters) are both close to a third, they cannot be too far apart from each other. This property allows us to infer potential relationships between concepts, as proximity to a common third concept suggests likely closeness. Such qualities make the VI metric a powerful tool for exploring complex relationships and predicting links within a network of concepts. For consistent comparisons between different pairs of concepts, it is essential to use the same underlying corpus and frequency domain. In particular, including zero occurrences $(k, m = 0)$ in the definitions

of $M(X,Y)$ and $H(X,Y)$ ensures that every document and possible frequency count is considered, preserving a common probability normalization. Otherwise, two pairs of concepts might both produce $d(X, Y) = 0$—one pair restricted to a handful of documents, the other spanning many more—yet the results would differ in their statistical significance if not measured on the same scale. By incorporating $(k, m = 0)$ as part of the distribution, we maintain a uniform basis to calculate $d(X,Y)$ between all pairs of concepts, allowing fair and interpretable comparisons within a document corpus of size $N$.

The frequency distributions of concepts are changing over time as new documents emerge. To account for this, let $P(x_k, t) = \frac{N_c(k,t)}{N(t)}$ be the time-dependent probability that a concept $c$ that appears in the $N_c(t)$ documents exactly $k$ times given a total corpus of size $N(t)$ selected over a period of time $t_0 + t$. Then the mutual information $M(X,Y,t)$ and the joint entropy $H(X,Y,t)$ become time dependent and can be written as [36,37]:

$$M(X, Y, t) = \sum_{k,m=0}^{\infty} P(x_k, y_m, t) \, \log_2\!\left(\frac{P(x_k, y_m, t)}{P(x_k, t) \, P(y_m, t)}\right), \tag{2}$$

$$H(X, Y, t) = -\sum_{k,m=0}^{\infty} P(x_k, y_m, t) \, \log_2\!\big(P(x_k, y_m, t)\big), \tag{3}$$

where $P(x_k, y_m, t)$ and $P(x_k, t) = \sum_{y_m} P(x_k, y_m, t)$ (respectively $P(y_m, t) = \sum_{x_k} P(x_k, y_m, t)$) represent the *joint* and *marginal* probability distributions of $X$ and $Y$ at time $t$, respectively.

Having the selected collection of concepts obtained from LDA or with expert-provided manual selection of the relevant collection of keywords that represents a topic of interest, we can calculate the pairwise symmetric distance matrix with elements $d(X_i, Y_j) \equiv d_{ij} = d_{ji}$ representing distances between the selected collection of concepts, which encapsulates information about the topological properties and complexity of the corresponding topic knowledge network. Our findings show that temporal changes in topic pairwise distances, $\Delta d/\Delta t$ driven by changes in the number of relevant documents for each pair of concepts expressed in terms of related velocity matrices, allow a granular study of the temporal evolution of the network.

The relative velocity matrix is derived by taking the differences between the elements of the distance matrices at various times $t_n$, where $n \in \mathbb{Z}$. Using a fixed time interval $\Delta t = t_{n+1} - t_n$ of one year, noise and computational demands are minimized. For ease of notation, we denote $\Delta t$ by 1. To maintain comparability across varying scales of distance values, we suggest to compute the relative symmetric difference as the following ratio for each entry in the velocity matrix:

$$v_{ij} = \frac{d_{ij}^{t_{n+1}} - d_{ij}^{t_n}}{d_{ij}^{t_{n+1}} + d_{ij}^{t_n}}. \tag{4}$$

The resulting relative "velocity" matrix may contain both positive and negative elements, reflecting the increasing or decreasing proportional weight of documents that mention both concepts across successive time slices. A positive value indicates that the relative occurrence of co-mentions is decreasing, leading to an increasing conceptual distance, while a negative value suggests that co-mentions are becoming more frequent, reducing the distance between concepts. In other words, if the relative number of such documents increases, the conceptual distance decreases, indicating stronger topic convergence. Conversely, if the relative number of co-mentions declines, the conceptual distance grows, signifying topic divergence.

We represent the velocity matrix as the sum of two components, each containing elements of the same sign: one corresponding to increasing proximity (negative values) and the other

to increasing divergence (positive values). By computing a Minimum Spanning Tree (MST) separately for these two components, we can identify the most dynamic and influential connections within the network of concepts at each time step. The smallest negative component extracted via MST highlights the strongest conceptual convergences, where concepts become more closely related, while the largest positive components reveal divergent subtopics whose components are moving further apart.

In the following section, we apply our model to analyze the dynamics of the topics related to International Relations discipline.

## Results and discussion

To analyze the evolution of the topic within the conceptual network, we first extract a subset of concepts closely related to a given seed concept based on the normalized Variation of Information (NVI) metric, considering all available documents in our data set. This subset is then defined as a *topic*, named after the seed concept.

In this approach, the topic size is determined by the degree centrality of the seed concept within the entire network, subject to a cutoff threshold applied to the distance measure between the seed concept and all other concepts in the network. Since most seed concepts have a relatively limited number of direct associations compared to the entire network, this method enables a more focused and computationally efficient analysis while preserving meaningful conceptual relationships.

In practice, the collection of concepts associated with a given topic is constructed through a systematic filtering procedure. The selection process involves three key steps:

**Step 1: Define the Relevant Document Set**

- Iterate through the document corpus.
- Identify and select documents where the **seed concept** appears at least $k$ times ($k>1$), where $k$ is a frequency threshold for the term.
- Store these selected documents as the **document set** $D$.

**Step 2: Identify Concept Candidates**

- Within **document set** $D$, extract all distinct **occurring concepts**.
- Retain only those concepts that appear in at least in $n$% of documents in $D$, ensuring statistical significance.
- Store these selected terms in a **concept set** $C$.

**Step 3: Filter Concepts by Distance to the Seed Concept**

- Compute the **NVI distance** $d(\text{Seed Concept}, c_j)$ for each candidate concept $c_j$ in $C$.
- Apply a distance threshold $b$, retaining only those concepts with $d<b$.
- The resulting filtered set represents the **topic-specific conceptual network**.

The final selection process ensures that only concepts with strong thematic and statistical relevance to the seed concept are included in the analysis. This filtered subset is used to study topic evolution, structural properties, and interaction patterns within the conceptual network.

It is worth noting that topic concepts can be selected using various alternative approaches, as discussed in the Introduction. For example, LDA or other topic modeling techniques can be employed to extract relevant concepts in a data-driven manner. Additionally, expert-driven methods, such as domain-specific taxonomies or manual curation, offer another

**Table 1. Selected seed concepts, related concepts, and protocol parameters.**

| Seed Concept | Related Concepts | Protocol Parameters |
|---|---|---|
| international security | international relations, international affairs, security, terrorism, security council, national security, weapons, securitization, afghanistan, threats, defence, insecurity, terrorist attacks, attacks, conflicts, russia, combat, cooperation, cornell university, terror, allies | $n = 5\%$, $b = 0.984$ |
| international relations | security, security dialogue, securitization, diplomacy, politics, cooperation, terrorism, dialogue, national security, borders, cornell university, defence, treaty, russia, human rights, regimes, agreements, threats, democracy, conflicts | $n = 5\%$, $b = 0.982$ |
| sanctions | security council, compliance, russia, weapons, courts, violation, security, national security, president, provisions, treaty, human rights, crimes, legislation, cooperation, congress, officials, agreements, defence, detention, obligations, terrorism | $n = 5\%$, $b = 0.982$ |
| climate change | global environmental, emissions, global climate, intergovernmental panel, global warming, mitigation, ecosystem, adaptation, sustainability, conservation, disasters, temperature, pollution, disaster, sustainable development, ecology, scientists, livelihoods, agriculture | $n = 5\%$, $b = 0.96$ |
| cybersecurity | cybercrime, malware, cyberspace, hackers, critical infrastructure, artificial intelligence, external security, internal security, counterterrorism, microsoft, regulators, security threats, privacy, computer, vulnerabilities, countering, terrorism, decisionmaking, homeland security, oversight | $n = 5\%$, $b = 0.991$ |
| human rights | protection, refugee, detention, justice, security, violence, obligations, borders, trafficking, asylum seekers, violation, courts, victims, amnesty, high commissioner, migration, migrants, immigrants, government, persecution | $n = 5\%$, $b = 0.968$ |

The table presents selected seed concepts, their corresponding related concepts, and the protocol parameters used for concept selection based on document occurrence threshold ($n\%$), and NVI distance cutoff ($b$).

way to define topic boundaries. However, since the primary objective of this study is not to develop a robust topic extraction method that is comparable in efficiency to LDA or expert-based approaches, we adopted a simplified filtering protocol. This protocol allows us to select concepts based on their co-occurrence structure and study the dynamics within the selected cloud of concepts, focusing on their evolving relationships over time.

Table 1 presents the seed concepts along with the top 20 related concepts extracted at the frequency threshold $k = 2$ for the seed concept, identified based on their distance NVI from the seed concept in the data set at 2024. Furthermore, we specify the protocol parameters used for the selection of concepts, including the document occurrence threshold ($n\%$) and the distance cutoff ($b$), which define the inclusion criteria for related concepts.

To ensure that all concepts are clearly visible and their relationships traceable in the figures, we adjust the threshold $b$ to select only the top 20 most relevant concepts. Although including more concepts could provide a broader view of the network, it would also reduce the readability of the figures. For a more comprehensive perspective, we provide a Mathematica notebook to obtain plots with an extended set of related concepts in the supplementary materials (see Data Availability section).

## Concept contextual evolution in cybersecurity topic

In Fig 1, we present a *bubble-flow graph* illustrating the *converging* (red) and *diverging* (blue) connections between concepts related to the concept of 'cybersecurity' as a seed concept in our topic selection protocol. This visualization captures the *dynamical evolution* of the conceptual landscape, allowing us to track how key concepts emerge, strengthen, or weaken over time.

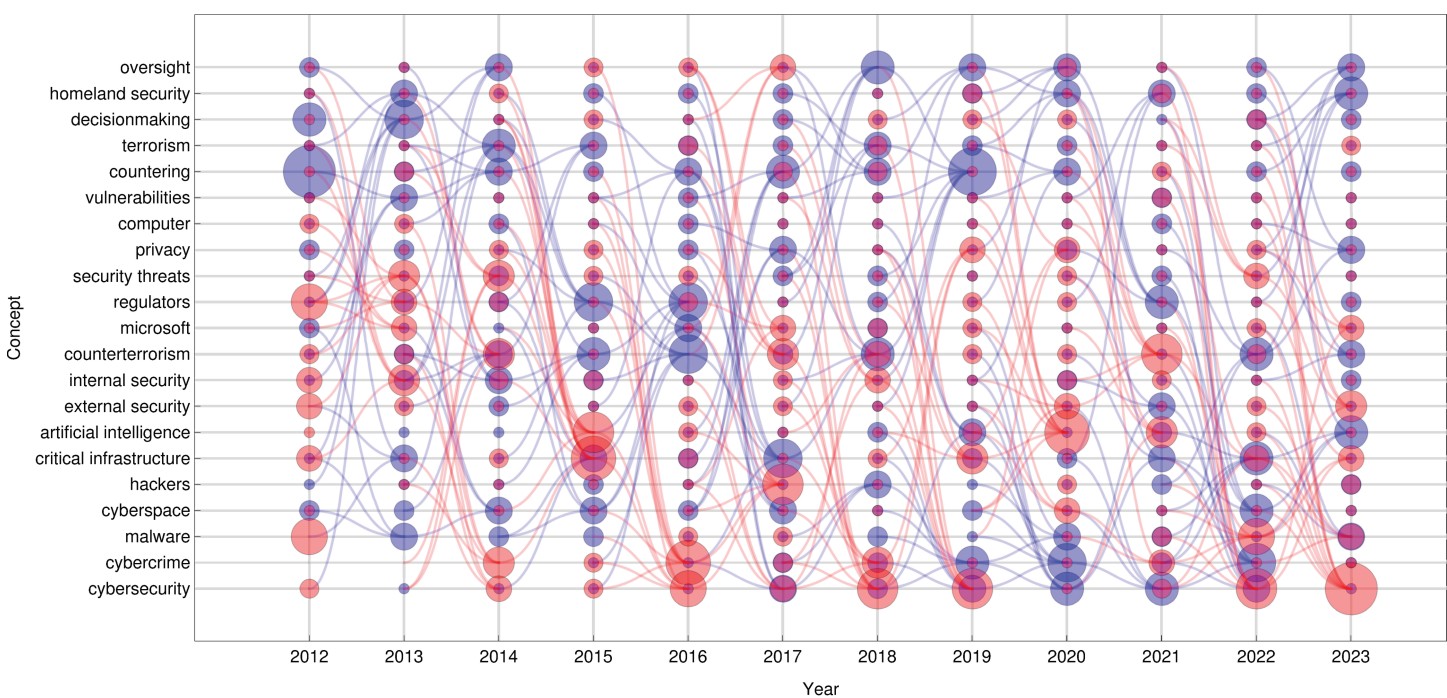

**Fig 1. Concept Evolution in Cybersecurity: Converging and Diverging Topics Over Time.** The bubble-flow graph illustrates the evolution of topics related to the cybersecurity seed concept. The visualization highlights topic convergence (red curves and bubbles), where concepts move closer to a *hub* concept in terms of the *d* metric, and topic divergence (blue curves and bubbles), where concepts become less related to the hub concept over time. The size of the bubbles reflects the degree centrality of each node in the MST-based topic network.

To construct this representation, we compute the velocity matrix at each yearly time step, which quantifies the changes in conceptual distances between concepts related to cybersecurity over consecutive years. The velocity matrix is then decomposed into its positive and negative components, corresponding to concepts that move closer together (converging) and those that move further apart (diverging).

To extract the most structurally significant conceptual shifts, we apply the MST algorithm separately to the positive and negative components of the velocity matrix at each time slice. This allows us to identify the most influential conceptual transitions, filter out minor fluctuations, and highlight the dominant patterns of conceptual change.

The bubble sizes in the graph represent the degree centrality of each concept within the MST, indicating its relative influence on the shaping of the cybersecurity domain at each time step. The flowing edges, represented as smooth Bézier curves, illustrate the continuity or disruption of conceptual relationships over time.

- **Red curves** indicate *intensifying connections*, suggesting the emergence or reinforcement of conceptual ties.
- **Blue curves** denote *weakening relationships*, capturing the divergence or fading influence of concepts over successive years.

This visualization enables us to distinguish between *stable, long-term conceptual anchors* and *ephemeral trends*, offering insight into the mechanisms driving:

- **Topic convergence** – where previously distinct concepts become more closely linked.

- **Topic divergence** – where formerly related concepts gradually lose their association.

The temporal convergence of concepts toward 'cybersecurity' reveals distinct waves of thematic realignment that correspond to major geopolitical, technological, and regulatory events. Between 2013 and 2014, the variation of information distance significantly decreased between 'cybersecurity' and terms such as 'critical infrastructure' and 'counterterrorism', reflecting heightened concern about national resilience in the wake of the Snowden revelations [39] and renewed discourse on infrastructure protection after Stuxnet [40]. In 2015–2016, the focus shifts to 'cybercrime', 'malware' and 'artificial intelligence', reflecting the attention of the OPM data breach [41], Sony Pictures hack [42], and concerns over the potential for AI to be used in disinformation campaigns during the 2016 US election [43]. The 2016–2017 period is marked by a convergence with 'hackers' and 'external security', which aligns with the rise of nation-state cyber threats [44]. By 2017–2018, the conceptual space expands to include 'cyberspace', 'privacy', and 'security threats', consistent with WannaCry [45] and NotPetya [46] ransomware outbreaks and the global push to implement the General Data Protection Regulation (GDPR) of the EU [47]. The 2018–2019 window reinforces the role of cybersecurity in internal security and antiterrorism strategies, amid ongoing concern about online radicalization and surveillance infrastructure [48]. The 2021–2022 period reflects a shift toward institutional governance, with a growing emphasis on regulators and infrastructure protection following the Colonial Pipeline ransomware attack [49] and the US Executive Order to Improve Nation's Cybersecurity [50]. Most recently, in 2022-2023, we observed convergence with "Microsoft", "AI" and "cyberspace," highlighting the mainstreaming of generative AI technologies (e.g., ChatGPT) [51,52], increased private sector cybersecurity partnerships [53], and the dimension of cyber warfare of the Russia-Ukraine conflict [54].

These patterns are further reinforced by the observation that 'artificial intelligence' itself emerges as a local hub of conceptual convergence in 2014–2015 and again in 2019–2020. During 2014–2015, its proximity to terms such as 'cyberspace', 'privacy' and 'decision making' anticipates the growing discourse on AI's role in surveillance and algorithmic governance. By 2019–2020 and later, its convergence with 'critical infrastructure', 'homeland security' and 'oversight' reflects increasing regulatory scrutiny and recognition of AI's strategic importance in both public-sector security and civil society. These early and intermediate signals strengthen the conclusion that AI has been an evolving point of integration across cybersecurity, governance, and geopolitics - well before its mainstream visibility in 2022–2023.

The degree order of the nodes, as illustrated in Fig 2, demonstrates a periodic increase and decrease in the contextual proximity of both 'cybersecurity' and 'artificial intelligence' to other concepts within this topic. These fluctuations suggest cycles of integration and differentiation, where new research areas periodically become highly relevant before stabilizing or branching into distinct subfields. Such structural changes in the network highlight how cybersecurity continuously evolves, incorporating new paradigms, challenges, and interdisciplinary influences over time.

The different sizes of the data set between consecutive time slices pose a potential challenge, particularly when the data set expands while specific target concepts, for which we measure distance, are not mentioned in newly added documents. As shown in S1 Appendix, in such cases, the distance $d$ exhibits a systematic decrease over time due to the increasing dominance of probability $P(0,0)$ in Eqs (2) and (3). This effect may bias the estimation of mutual information, making it appear artificially large, as the absence of concepts in newer documents skews the underlying probability distribution. However, as demonstrated in Appendix A, the most significant contribution to $d$ originates from documents in which concepts co-occur contributing to the joint probabilities $P(k>0,m>0)$. The influence of these

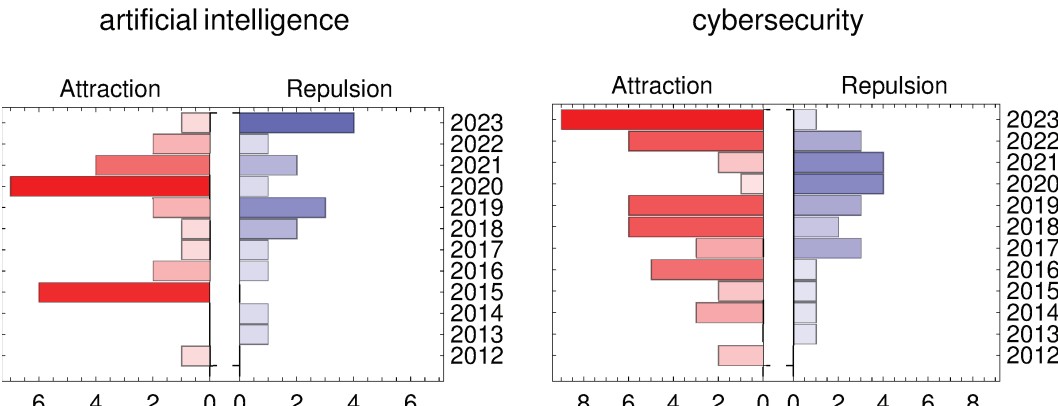

**Fig 2. Node order degree evolution for 'cybersecurity' and 'artificial intelligence' concepts.** Evolution of 'cybersecurity' and 'artificial intelligence' node order degrees over 2012-2023 in MST knowledge network related to 'cybersecurity' topic.

*relevant* documents represents the *fast* changing component that contributes to the dynamics of the NVI distance $d$ compared to the slow and predictable contribution of the background documents.

To ensure that our analysis captures the most significant structural changes, we apply the Minimum Spanning Tree (MST) algorithm to filter meaningful distance variations. This approach assumes that the dominant contributions to MST-selected distances originate primarily from relevant documents, while the slow accumulation of background documents (which do not mention target concepts) has a negligible effect on the network's core structure. Additionally, reducing the time step used in the analysis, can further minimize the slow-trend bias, particularly when the dataset grows rapidly. In such cases, finer temporal granularity can be advisable to better account for dataset expansion effects. Finally, the growth dynamics of the observed dataset can inform strategies for normalizing the distance $d$, allowing us to refine the methodology for different dataset size distributions over time.

## Conclusion

This study examines the structural evolution of topics in International Security through an information-theoretic lens, providing empirical evidence of how concepts such as cybersecurity, climate change, sanctions, human rights, and international security have evolved over time. By analyzing temporal contextual proximity, we identify recurring patterns of conceptual convergence and divergence, revealing how knowledge structures adapt to emerging research agendas and geopolitical developments. Our analysis demonstrates that thematic clusters form, dissolve, and reorganize dynamically, reflecting broader transformations in policy discourse, academic priorities, and international relations scholarship.

Our findings confirm that dataset expansion alone does not significantly influence conceptual distance. Instead, conceptual proximity is primarily dictated by co-occurrence patterns, the frequency with which two concepts appear together in documents, rather than the sheer size of the dataset. The observed relationships between mutual information, joint entropy, and document co-occurrence frequencies reinforce this conclusion, underscoring the importance of relational dynamics over corpus size in shaping conceptual structures. By applying velocity matrices and the Minimum Spanning Tree (MST) algorithm, we quantify how concepts dynamically cluster, fragment, and restructure, offering a fine-grained quantitative perspective on knowledge evolution within the field of International Relations.

The methodological framework presented in this study offers a systematic approach to tracking topic evolution in scientific research. Future research could expand upon this work by integrating external geopolitical and regulatory factors to assess their impact on topic dynamics. Furthermore, incorporating AI-driven predictive modeling could enhance the ability to anticipate topic trajectories, enabling the early identification of emerging research trends.

Our findings contribute to a broader understanding of the self-organizing nature of international relations research, demonstrating how knowledge structures adapt and evolve in response to technological advancements, policy transformations, and global challenges. As interrelated issues such as cybersecurity, human rights, and climate change continue to shape international discourse, tracking their structural evolution will be crucial for researchers, policymakers, and practitioners seeking to navigate and respond to complex geopolitical and academic landscapes.

## Supporting information

**S1 Appendix. Correction to distances from changing data set size.**
(PDF)

**S1 Fig. Dependence of the Conceptual Distance $d(N)$ on Dataset Size for Different Document Removal Strategies.** For two target concept pairs – "Pair 1" (blue palette) and "Pair 2" (red palette) – the distance $d(N)$ is calculated over a range of data set sizes from $N = 10\,000$ to $N = 100\,000$, with $N$ increasing in steps of $1\,000$. In these experiments, only the number of documents with zero citation frequencies for the target pair is increased. For each pair, three series of $d(N)$ data points are calculated and fitted with the model in Eq (5), corresponding to the following scenarios:

1. **Baseline:** All relevant documents (that is, those with $k,m>0$) are retained, with the baseline data shown as empty squares and the fitted model plotted as a dashed curve.
2. **20% Removal:** At each step, 20% of the relevant documents are randomly removed from the union of all papers that mention at least one of the target concepts, while the background (documents with zero citations for the target pair) increases by $1\,000$.
3. **50% Removal:** Similarly, 50% of the relevant documents are randomly removed at each step (green points for Pair 1 and orange points for Pair 2).

Each subfigure represents a different target for random document removal: subfigure (a) shows removal from the union of all papers mentioning either concept; subfigure (b) shows removal from the intersection (i.e. only those documents that mention both concepts); and subfigures (c) and (d) show removal from the complement (i.e. only those documents that mention a single concept from the target pair).

To demonstrate that the dynamics of the conceptual distance $d$ for a given pair of concepts $A$ and $B$ is weakly influenced by the size of the data set $N$ and the increasing number of background documents with zero concept frequencies, and that the primary driver of the decrease in conceptual distance is the number of documents mentioning both concepts $N_{AB}$, we present in S2 Fig the time series for key metrics observed for multiple pairs of concepts in the studied dataset. Key metrics include NVI distance $d$, mutual information, joint entropy, total size of the dataset $N$, number of background documents $N_{00}$, the ratio of documents mentioning both concepts $N_{AB}$ to the total size of the dataset, and the ratio of $N_{AB}$ to the sum of documents $N_A + N_B$ mentioning only one concept $A$ or $B$.
(TIF)

**S2 Fig. Dynamics of conceptual distance *d* and the growth effects of the data set** Time series of key metrics that illustrate the relationship between conceptual distance *d*, the size of the data set *N*, relevant documents and the number of background documents $N_{00}$. The plots present the dynamics of the NVI distance *d* between concepts *A* and *B*, their mutual information $M(A,B)$, joint entropy $H(A,B)$, total dataset size *N*, the number of background documents $N_{00}$ (documents that do not mention these concepts), the ratio of documents mentioning both concepts $N_{AB}$ to the total dataset size, and the ratio of $N_{AB}$ to the sum of documents mentioning only one of the concepts, $N_A + N_B$. The results indicate that while dataset expansion alters global entropy-based measures, the conceptual distance is predominantly shaped by the number of co-occurring mentions of the concepts, underscoring the role of shared contextual usage rather than dataset size alone.

The analysis of conceptual distance dynamics highlights that the expansion of the dataset, as reflected in the dynamics of *N* or $N_{00}$ in S2 Fig, does not affect the behavior of the distance *d* in the manner predicted by the model in Eq (5) when only the number of background documents of type $N_{00}$ increases. This finding suggests that the conceptual distance is primarily dependent on the number of documents in which the target concepts co-occur, which is evident from the behavior of the ratios $N_{AB}/N$ and $N_{AB}/(N_A + N_B)$: as these ratios increase, the distance *d* decreases.

Furthermore, the effect of random variations on the number of relevant documents, as observed in S1 Fig, reinforces this conclusion. The random removal of relevant documents affects the dynamics of *d* in a way that is consistent with its dependence on the presence of concept co-occurrence, rather than merely on dataset size and the number of background documents. These results collectively indicate that meaningful conceptual proximity is driven by shared document contexts rather than by the absolute number of documents in the dataset. (TIF)

**S3 Fig. Evolution of the semantic proximity between concepts in the international relations topic.** The bubble-flow graph illustrates the evolution of topics related to the 'International Relations' seed concept. The visualization highlights *topic convergence* (red curves and bubbles), where concepts move closer to a *hub* concept in terms of the *d* metric, and *topic divergence* (blue curves and bubbles), where concepts become less related to the hub concept over time. The size of the bubbles reflects the degree centrality of each node in the MST-based topic network. (TIF)

**S4 Fig. Evolution of the semantic proximity between concepts on the topic of climate change: Converging and Diverging Topics Over Time.** The bubble-flow graph illustrates the evolution of topics related to the 'Climate change' seed concept. The visualization highlights *topic convergence* (red curves and bubbles), where concepts move closer to a *hub* concept in terms of the *d* metric, and *topic divergence* (blue curves and bubbles), where concepts become less related to the hub concept over time. The size of the bubbles reflects the degree centrality of each node in the MST-based topic network. (TIF)

**S5 Fig. Evolution of the semantic proximity between concepts on the topic of international security: Converging and Diverging Topics Over Time.** The bubble-flow graph illustrates the evolution of topics related to the 'International Security' seed concept. The visualization highlights *topic convergence* (red curves and bubbles), where concepts move closer to a *hub* concept in terms of the *d* metric, and *topic divergence* (blue curves and bubbles), where

concepts become less related to the hub concept over time. The size of the bubbles reflects the degree centrality of each node in the MST-based topic network.

(TIF)

**S6 Fig. Evolution of the semantic proximity between concepts in the sunctions topic: Converging and Diverging Topics Over Time.** The bubble-flow graph illustrates the evolution of topics related to the 'Sunctions' seed concept. The visualization highlights *topic convergence* (red curves and bubbles), where concepts move closer to a *hub* concept in terms of the $d$ metric, and *topic divergence* (blue curves and bubbles), where concepts become less related to the hub concept over time. The size of the bubbles reflects the degree centrality of each node in the MST-based topic network.

(TIF)

**S7 Fig. Evolution of the semantic proximity between concepts in the security topic: Converging and Diverging Topics Over Time.** The bubble-flow graph illustrates the evolution of topics related to the 'Security' seed concept. The visualization highlights *topic convergence* (red curves and bubbles), where concepts move closer to a *hub* concept in terms of the $d$ metric, and *topic divergence* (blue curves and bubbles), where concepts become less related to the hub concept over time. The size of the bubbles reflects the degree centrality of each node in the MST-based topic network.

(TIF)

## Acknowledgments

The author thanks Alexander Yakimenko for insightful discussions on the NVI methodology and its application to topic dynamics. The author also acknowledges access to the 'International Security' corpus provided by the JSTOR Data for Research program. The author thanks the reviewers for their constructive comments and suggestions.

## Author contributions

**Conceptualization:** Artem Chumachenko.

**Data curation:** Artem Chumachenko.

**Formal analysis:** Artem Chumachenko.

**Investigation:** Artem Chumachenko.

**Methodology:** Artem Chumachenko.

**Software:** Artem Chumachenko.

**Visualization:** Artem Chumachenko.

**Writing – original draft:** Artem Chumachenko.

**Writing – review & editing:** Artem Chumachenko.

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
