## [Decision Letter · Decision Letter 0]

16 May 2025

PONE-D-25-11688Quantifying hot topic dynamics in scientific literature: an information-theoretical approachPLOS ONE

Dear Dr. Chumachenko,

Thank you for submitting your manuscript to PLOS ONE. After careful consideration, we feel that it has merit but does not fully meet PLOS ONE’s publication criteria as it currently stands. Therefore, we invite you to submit a revised version of the manuscript that addresses the points raised during the review process. In your revision, please take special attention to the state-of-the-art research and the question of open data.

We look forward to receiving your revised manuscript.

Kind regards,

Malte Rehbein

Academic Editor

PLOS ONE

Journal Requirements:

3. We are unable to open your Supporting Information file and Figure file S1 Fig to S7 Fig and Fig 1, Fig2. Please kindly revise as necessary and re-upload.

Reviewers' comments:

Reviewer's Responses to Questions

**Comments to the Author**

1. Is the manuscript technically sound, and do the data support the conclusions?

Reviewer #1: Yes

Reviewer #2: Yes

2. Has the statistical analysis been performed appropriately and rigorously? 

Reviewer #1: Yes

Reviewer #2: Yes

3. Have the authors made all data underlying the findings in their manuscript fully available?

Reviewer #1: Yes

Reviewer #2: Yes

4. Is the manuscript presented in an intelligible fashion and written in standard English?

Reviewer #1: Yes

Reviewer #2: Yes

5. Review Comments to the Author

Reviewer #1: 1- Please explain the researcher's conclusions and what he addressed within the research gap of the work presented in the abstract.

2- Ensure that references are included for all equations in the research.

3- The research paper lacks tables and charts that reflect the researcher's reading and summary of the reviewed research, and does not highlight the strengths and weaknesses of previous research.

4- Add references published in 2025

Reviewer #2: The main objective of the study is to analyse the dynamics of research topics and the conceptual dependencies within the scientific discourse. This approach is presented as an alternative to traditional topic modeling methods (such as DTM or neural topic models). These either have difficulties in capturing dynamic relationships, require high computational resources or are based on weighting methods without metric properties. The core of the article is a complex and promising methodological development. It is not about an empirical study and the presentation of new empirical findings. Therefore, an explicit research question and a research design are also missing, which is understandable given the focus of the article. The methodological approach consists of constructing conceptual networks based on a mathematically founded distance metric (NVI). The temporal development of these distances is quantified using velocity matrices. The MST algorithm is used to identify the most important dynamic connections (convergence and divergence) in order to gain insights into the development of knowledge structures and to visualize them.

Overall, the article is very well written and sufficiently illustrates the complex methodological approach. The limitations of the work are also explained in detail. In my view, there are only three minor weaknesses, but these can be rectified relatively quickly.

1. Although the current state of research on topic modeling is discussed, important studies are missing. For example, the ‘Structural Topic Modeling (STM)’ developed by Roberts et al. (2014) is not taken into account, although it is used in science studies, sociology, political science and many other disciplines. The STM and exemplary studies should also be taken up and critically evaluated.

2. The temporal development of the topics, i.e. the results of the study, could be presented in a more differentiated way in some places and linked back to existing specialised discourses. This would facilitate the integration of the results within the existing research discourse, thereby enhancing the visibility of the added value of the novel methodological approach.

3. Although it is mentioned that the code is available on GitHub, there is no link to this page. My own research was also unsuccessful. The article should contain a GitHub link. The code should be sufficiently documented on GitHub. It would also be in the interest of the authors if their code could provide a basis for further studies.

6. PLOS authors have the option to publish the peer review history of their article (what does this mean?). If published, this will include your full peer review and any attached files.

Reviewer #1: No

Reviewer #2: **Yes: **Markus Eckl

---

## [Author Response · Author response to Decision Letter 1]

2 Jun 2025

Response to Reviewers and Editor

We sincerely thank the reviewers and the editor for their constructive comments and valuable suggestions, which helped improve the quality and clarity of the manuscript.

Editor Comments

1. Please ensure that your manuscript meets PLOS ONE’s style requirements, including those for file naming.

Response:

All formatting and file naming requirements have been verified and are now fully satisfied.

2. The grant information provided in the ‘Funding Information’ and ‘Financial Disclosure’ sections do not match. Please ensure consistency.

Response:

The following unified statement is now used in both sections:

“The author received no specific funding for this work.”

3. We are unable to open your Supporting Information and figure files (S1 Fig to S7 Fig and Fig 1, Fig 2). Please revise and re-upload.

Response:

All supplementary files are provided in `.eps` format. I have verified that the files are not corrupted and can be opened using online EPS viewers (e.g., https://jumpshare.com/viewer/eps). Figures Fig 1 and Fig 2 are also available at the end of the submission file in EPS format for visual inspection.

Reviewer #1 Comments

1. “Please explain the researcher’s conclusions and what he addressed within the research gap of the work presented in the abstract.”

Response:

The abstract has been revised to clearly state the research gap, the proposed solution, and the main findings.

2. “Ensure that references are included for all equations in the research.”

Response:

References have been added before the introduction of formulas (1), (2), and (3). Formulas (4) and (5) are original contributions by the author and are noted as such.

3. “The research paper lacks tables and charts that reflect the researcher’s reading and summary of the reviewed research, and does not highlight the strengths and weaknesses of previous research.”

Response:

We appreciate this valuable comment. While comparative tables are useful in methodological surveys, our manuscript is not intended as a review or benchmark of topic detection methods. Rather, our contribution is a complementary approach that focuses on within-topic structural dynamics and conceptual proximity analysis, which are not the primary targets of conventional topic modeling techniques (e.g., LDA, DTM, NTMs).

Our method can be applied in conjunction with topic models. For example, once a topic is identified via LDA, our framework can track semantic drift, detect convergence hubs, and evaluate the influence of knowledge producers on concept evolution. To clarify this distinction, we have added a paragraph to the Introduction section.

4. “Add references published in 2025.”

Response:

We have added a recent reference: Tan Z, D’Souza J. Bridging the Evaluation Gap: Leveraging Large Language Models for Topic Model Evaluation. arXiv preprint. 2025; arXiv:2502.07352. It is cited in the Introduction section.

Reviewer #2 Comments

1. “The current state of research on topic modeling is discussed, but important studies such as Structural Topic Modeling (STM) by Roberts et al. (2014) are missing.”

Response:

We have added discussion of STM and relevant citations to the Introduction section.

2. “The temporal development of topics should be linked back more clearly to specialized discourses.”

Response:

We have expanded the Results section with contextual interpretations for the observed evolution of “cybersecurity” and “artificial intelligence” as key concepts.

3. “No working link was found to the GitHub repository; please ensure the code is available and documented.”

Response:

The code and data are accessible via the Zenodo repository listed in the Data Availability section: https://doi.org/10.5281/zenodo.14948128, which includes a link to the GitHub repository: https://github.com/ArtemEsper/TopicGeometry. The repository is documented and includes the necessary scripts and usage instructions.

---

## [Decision Letter · Decision Letter 1]

23 Jun 2025

Quantifying hot topic dynamics in scientific literature: an information-theoretical approach

PONE-D-25-11688R1

Dear Dr. Chumachenko,

We’re pleased to inform you that your manuscript has been judged scientifically suitable for publication and will be formally accepted for publication once it meets all outstanding technical requirements.

Kind regards,

Malte Rehbein

Academic Editor

PLOS ONE

Additional Editor Comments (optional):

Reviewers' comments:

Reviewer's Responses to Questions

**Comments to the Author**

1. If the authors have adequately addressed your comments raised in a previous round of review and you feel that this manuscript is now acceptable for publication, you may indicate that here to bypass the “Comments to the Author” section, enter your conflict of interest statement in the “Confidential to Editor” section, and submit your "Accept" recommendation.

Reviewer #1: (No Response)

Reviewer #2: All comments have been addressed

2. Is the manuscript technically sound, and do the data support the conclusions?

Reviewer #1: (No Response)

Reviewer #2: Yes

3. Has the statistical analysis been performed appropriately and rigorously? 

Reviewer #1: (No Response)

Reviewer #2: Yes

4. Have the authors made all data underlying the findings in their manuscript fully available?

Reviewer #1: (No Response)

Reviewer #2: Yes

5. Is the manuscript presented in an intelligible fashion and written in standard English?

Reviewer #1: (No Response)

Reviewer #2: Yes

6. Review Comments to the Author

Reviewer #1: (No Response)

Reviewer #2: All points of criticism were taken up by the authors and dealt with in an adequate manner.

I wish you every success with the publication of your article.

7. PLOS authors have the option to publish the peer review history of their article (what does this mean?). If published, this will include your full peer review and any attached files.

Reviewer #1: No

Reviewer #2: **Yes: **Markus Eckl

---

## [Editor Report · Acceptance letter]

PONE-D-25-11688R1

PLOS ONE

Dear Dr. Chumachenko,

I'm pleased to inform you that your manuscript has been deemed suitable for publication in PLOS ONE. Congratulations! Your manuscript is now being handed over to our production team.

Kind regards,

on behalf of

Prof. Dr. Malte Rehbein

Academic Editor

PLOS ONE